# Intraintestinal Delivery of Tastants Using a Naso-Duodenal-Ileal Catheter Does Not Influence Food Intake or Satiety

**DOI:** 10.3390/nu11020472

**Published:** 2019-02-23

**Authors:** Tim Klaassen, Annick M. E. Alleleyn, Mark van Avesaat, Freddy J. Troost, Daniel Keszthelyi, Adrian A. M. Masclee

**Affiliations:** 1Division of Gastroenterology-Hepatology, Department of Internal Medicine, School of Nutrition and Translational Research in Metabolism, Maastricht University Medical Center+, P.O. Box 5800, 6202 AZ Maastricht, The Netherlands; a.alleleyn@maastrichtuniversity.nl (A.M.E.A.); mark.van.avesaat@mumc.nl (M.v.A.); f.troost@maastrichtuniversity.nl (F.J.T.); d.keszthelyi@mumc.nl (D.K.); a.masclee@mumc.nl (A.A.M.M.); 2Food Innovation and Health, Center for Healthy Eating and Food Innovation, Maastricht University, 5911 AA Venlo, The Netherlands

**Keywords:** satiety, tastants, food intake, intraduodenal infusion, intraileal infusion, overweight, weight management

## Abstract

Intraduodenal activity of taste receptors reduces food intake. Taste receptors are expressed throughout the entire gastrointestinal tract. Currently, there are no data available on the effects of distal taste receptor activation. In this study, we investigate the effect of intraduodenal and/or intraileal activation of taste receptors on food intake and satiety. In a single-blind randomized crossover trial, fourteen participants were intubated with a naso-duodenal-ileal catheter and received four infusion regimens: duodenal placebo and ileal placebo (DPIP), duodenal tastants and ileal placebo (DTIP), duodenal placebo and ileal tastants (DPIT), duodenal tastants and ileal tastants (DTIT). Fifteen minutes after cessation of infusion, subjects received an *ad libitum* meal to measure food intake. Visual analog scale scores for satiety feelings were collected at regular intervals. No differences in food intake were observed between the various interventions (DPIP: 786.6 ± 79.2 Kcal, DTIP: 803.3 ± 69.0 Kcal, DPIT: 814.7 ± 77.3 Kcal, DTIT: 834.8 ± 59.2 Kcal, *p* = 0.59). No differences in satiety feelings were observed. Intestinal infusion of tastants using a naso-duodenal-ileal catheter did not influence food intake or satiety feelings. Possibly, the burden of the four-day naso-duodenal-ileal intubation masked a small effect that tastants might have on food intake and satiety.

## 1. Introduction

Obesity is considered a major healthcare problem with worldwide obesity almost being tripled since 1975 [1]. Therefore, there is an increasing need for non-invasive therapies for weight management. Gastrointestinal (GI) hormones, such as cholecystokinin (CCK) and glucagon-like peptide-1 (GLP-1), have been shown to reduce food intake and hunger after intravenous administration [2,3,4]. Therefore, the GI-tract is an interesting target for non-invasive therapies to reduce food intake and induce satiety/satiation. 

Intestinal macronutrient infusion decreases food intake and induces the release of CCK, GLP-1, and peptide YY (PYY) [5]. This mechanism is commonly referred to as intestinal- or ileal brake [6,7]. A recent review proposed a proximal to the distal gradient in the small intestine, where a more profound effect on food intake can be found after distal compared to proximal macronutrient infusion [8]. Previous studies have demonstrated that besides macronutrients, substances referred to as tastants are able to activate certain taste receptors in the GI-tract which are coupled to enteroendocrine cells (EEC), and can trigger the release of satiety hormones (i.e., CCK, GLP-1, and PYY) [9,10,11,12,13]. These taste receptors can be found throughout the entire GI-tract. Expression levels for the various taste receptor differ throughout the gut. Table 1 gives a simplified visual representation of the relative expression of taste receptors throughout the human gut based on current literature [14,15,16,17].

In a recent study, van Avesaat et al. have shown that duodenal infusion of a combination of sweet, bitter, and umami tastants significantly decreased *ad libitum* meal intake, whilst increasing satiety and decreasing hunger feelings. These effects were not accompanied by changes in systemic levels of GLP-1, PYY, and CCK [18]. Up to now, no data are available on the effect of activation of taste receptors in the more distal small intestine. Since one of the functions of taste receptors in the gut is to sense food being present in the lumen, it should be investigated whether the beforementioned proximal to distal gradient found for the intestinal brake is operative for taste receptor activation.

Therefore, in the present study, we compared the effects of intraduodenal infusion versus intraileal infusion of a combination of tastants (sweet, bitter, and umami) on *ad libitum* food intake, satiation, and GI-complaints in healthy subjects. Since sweet and umami taste are sensed by various subtypes of the taste receptor family 1 (TAS1R) and bitter taste is sensed by the taste receptor family 2 (TAS2R), the combination will activate a wide range of taste receptors. We hypothesized that infusing tastants at both infusion sites (duodenum and ileum) will decrease food intake and increase satiation to the greatest extent when compared with infusion of placebo or single port infusion. Infusing in solely the duodenum or the ileum will also decrease food intake and increase satiation when compared to placebo, albeit to a lesser degree than infusing at both infusion sites simultaneously. Furthermore, we expect intraileal delivery of tastants will decrease food intake and increase satiation to a greater extent when compared with intraduodenal delivery of tastants.

## 2. Materials and Methods 

This study was approved by the Medical Ethics Committee of the Maastricht University Medical Center+ (MUMC+), Maastricht, The Netherlands, and performed in full accordance with the Declaration of Helsinki (latest amendment by the World Medic Association in 2013) and Dutch Regulations on Medical Research Involving Human Subjects (WMO, 1998). This study was registered in the US National Library of Medicine (http://www.clinicaltrials.gov, ID NCT03140930). All subjects gave written informed consent before screening.

### 2.1. Subjects

Healthy men and women were recruited by local advertisements. Inclusion criteria were age between 18 and 65 years, a body mass index (BMI) between 18 and 25 kg/m^2^, with a stable weight over the past six months (<5% body weight change). Exclusion criteria were gastrointestinal complaints, history of chronic or severe disease, use of medication influencing endpoints within 14 days prior to testing, administration of investigational drugs which interfere with this study, major abdominal surgery, dieting, pregnancy or lactation, excessive alcohol consumption (>20 alcoholic consumptions per week), smoking, weight <60 kg, non-tasters of sweet, bitter or umami stimuli, evidence of monosodium glutamate (MSG)-hypersensitivity.

Prior to testing, screening was performed where abovementioned inclusion and exclusion criteria were checked, and a taste perception test was performed. Subjects tasted quinine (0.5 mmol/L), Reb A (50 mmol/L), MSG (50 mmol/L), and tap water blindly and had to indicate their sense of taste. Subjects had to identify each taste correctly in order to be eligible for the study. Furthermore, their length and weight were measured to calculate their BMI. 

A sample size calculation was based on the difference in meal intake between duodenal infusion of a combination of tastants and duodenal infusion of placebo as reported by van Avesaat et al. [18]. Using a difference in means of 64 Kcal, a standard deviation of difference of 63, a power of 80%, and an alpha of 1.67%, a total number of 13 subjects were needed. An alpha of 1.67% was used to correct for multiple testing.

### 2.2. Study Design

In this single-blind randomized, placebo-controlled crossover study, subjects received the combination of tastants (sweet, bitter, and umami) and/or placebo (tap water) in the duodenum and/or the ileum for four consecutive test days. This results in four combinations which were infused on the various test days: duodenal placebo and ileal placebo (DPIP), duodenal tastants and ileal placebo (DTIP), duodenal placebo and ileal tastants (DPIT), duodenal tastants and ileal tastants (DTIT).

### 2.3. Catheter Positioning

A 305 cm long silicon 9-lumen (8-lumen, 1 balloon inflation channel, the outer diameter of 3.5 mm) custom-made naso-ileal reusable catheter (Dentsleeve International, Mui Scientific, Mississauga, Canada) was used for intubation. 

One day prior to the first test day, subjects arrived at 7:40 AM at the Maastricht University Medical Center+ (MUMC+) after an overnight fast. If preferred by the subject, local anesthesia of nasal mucosa using xylocaine (10% spray, AstraZeneca, Zoetermeer, The Netherlands) was applied. After placement of the catheter in the stomach, the catheter was guided through the pylorus and into the duodenum under intermittent fluoroscopic control. Progression of the catheter from duodenum to ileum was performed as described earlier [19]. Fluoroscopy was used to check the positioning of the catheter on the first and the last test day. Radio-opaque markers were added to the infusion ports on the catheter, which accounted for the determination of the catheter position. On all test days, intestinal fluid was sampled from various infusion ports, and pH was measured using pH strips (MColorpHast™, Merck, Darmstadt, Germany) in order to estimate the catheter positioning.

### 2.4. Preparation and Infusion of Tastants

The combination of three tastants was infused in the duodenum, the ileum, or both the duodenum and the ileum. In order to prevent side effects from occurring, 75% of acceptable daily intake (ADI) of these tastants was infused. 540 mg Rebaudioside A (Reb A, Stevija Natuurlijk, Drachten, The Netherlands), 75 mg Quinine (Arnold Suhr, Hilversum, The Netherlands), and 2 g Monosodium Glutamate (MSG, Ajinomoto, Hamburg, Germany) were dissolved in 120 mL tap water and was used as tastant mixture for infusion, as was done by van Avesaat et al. [18]. All tastants used were non-caloric and yielded no nutritional value. The placebo infusion consisted of 120 mL of tap water. A magnetic stirrer was used to dissolve the tastants. The mixture was infused over a 60-min period with an infusion rate of 2 mL/min. This was consistent with the infusion rate of van Avesaat et al. mimicking the slow influx from the stomach to duodenum and slow transit through the gut in the ileum.

### 2.5. Protocol

On each test day, after an 8 h overnight fast, subjects arrived at 8:00 AM at the MUMC+. Subjects were instructed to consume the same habitual meal on the evening prior to testing. Hereafter, at t = 0 min, a standardized liquid breakfast meal (250 mL Goedemorgen drinkontbijt (Vifit); energy 145 Kcal per portion, 20.25 g carbohydrates, 8.5 g protein, and 2 g fat) was consumed. One hundred and fifty min (at t = 150 min) after breakfast consumption, a syringe containing the mixture for infusion was connected to the duodenal and ileal infusion port. The infusion was performed in 60 min at an infusion rate of 2 mL/min. Subjects received a standardized *ad libitum* lunch meal (Lasagna Bolognese (Plus supermarket); energy density per 100 g: 152 Kcal, 11 g carbohydrates, 7.1 g protein, and 8.6 g fat) fifteen min (at t = 225 min) after cessation of the infusion. The test meal was offered in excess and subjects were instructed to eat until they felt satiated. 

### 2.6. VAS for Satiation and GI-Complaints

Feelings of satiation-/satiety feelings and GI-complaints (e.g., satiety, hunger, stomach pain, and nausea) were measured using visual analog scales (VAS, 0–100 mm) scores at various time points (t = −30, 30, 90, 150, 165, 180, 195, 210, and 240 min) during the day. Subjects were asked to indicate on a line, anchored at the low end with the lowest intensity feelings, with opposing terms at the high end, which place on the scale best reflected their feeling at that moment [20]. 

### 2.7. Statistical Analyses

Data were analyzed using IBM SPSS statistics 24 (IBM Corporation, Armonk, NY, USA). A visual check of the normality of the data was performed. The primary outcome of this study was the amount of food intake in Kcal during an *ad libitum* lunch meal. Secondary outcomes were VAS scores for satiation-/satiety feelings and GI-complaints. 

Age, BMI, and gender were calculated by descriptive statistics. Food intake in Kcal and area under the curve (AUC) for VAS scores were compared using a linear mixed model with intervention (DTIP, DPIT, and DTIT, and DPIP), test day and the interaction of intervention × test day as fixed factors. When no significant interaction was found, the interaction was removed from the model to get the best model fit.

For VAS scores, a linear mixed model that included abovementioned fixed factors with the addition of fixed factors time and time × treatment interaction was also performed.

Data are presented as mean ± standard error of the mean (SEM) (unless specified otherwise), and a *p* < 0.05 was considered statistically significant.

## 3. Results

### 3.1. Subjects

In total, 19 subjects met the inclusion and exclusion criteria. Two subjects dropped out due to discomfort induced by the naso-ileal catheter, two subjects dropped out due to incorrect position of the catheter on the first test day, and one subject was excluded after not properly following the instructions for the *ad libitum* meal on the first test day. Therefore, 14 healthy volunteers (11 female, age 25.6 ± 10.5 years, BMI 22.3 ± 1.7 kg/m^2^) completed the study protocol and were included in the analyses.

### 3.2. Food Intake

No intervention × test day interaction was found. No differences in *ad libitum* food intake in Kcal were observed after intraduodenal, intraileal or combined infusion of tastants versus placebo infusion (DPIP: 786.6 ± 79.2 Kcal, DTIP: 803.3 ± 69.0 Kcal, DPIT: 814.7 ± 77.3 Kcal, DTIT: 834.8 ± 59.2 Kcal; *p* = 0.59) (Figure 1). Furthermore, as depicted in Figure 2, no trends in individual responses were found.

### 3.3. Satiation/Satiety Scores

The mean VAS scores for the desire to eat, hunger, satiety, and fullness are depicted in Figure 3. No differences in area under the curve (AUC_150–210_) for these VAS scores were observed between the various interventions. Furthermore, no intervention × timepoint interactions were found for these VAS scores. 

### 3.4. GI-Complaints

The mean VAS scores for stomach pain, bloating, and nausea are depicted in Figure 4. No differences in area under the curve (AUC_150–210_) for these VAS scores were observed between the various interventions. Furthermore, no intervention × timepoint interactions were found for these VAS scores. 

## 4. Discussion

Our results do not reveal any difference in satiety or food intake between duodenal administration, ileal administration or combined duodenal administration of a tastant mixture (sweet, bitter, and umami) or infusion of placebo. Moreover, no GI-complaints were caused by infusing tastants or placebo into the duodenum and/or the ileum.

Van Avesaat et al. have investigated the effect of intraduodenal infusion of the same tastant mixture on food intake [18]. In that study, intraduodenal infusion of this combination of tastants, in similar study design, using the same amount of tastants significantly reduced food intake by 64 Kcal and was accompanied by changes in satiation/satiety feelings. However, it must be noted that this is a small difference, which on its own might not be clinically significant. Repeating this effect multiple times per day with each meal might result in a clinically significant decrease of caloric intake. This difference in results of food intake between the two studies may be related to differences in study design. In the study of van Avesaat et al., the subjects were intubated with a naso-duodenal catheter on every test day for the administration of tastants. The catheter was removed immediately thereafter before the subjects were presented with the *ad libitum* test meal. In the present study, subjects were intubated for several days with a naso-ileal catheter, and therefore this catheter was present while meals were offered and ingested. We hypothesize that having a naso-ileal catheter in situ for multiple days negatively influences meal ingestion to such a degree that this masks the smaller magnitude of effect that infusion of non-caloric tastants into the intestine has. On the other hand, mean caloric intake showed no major differences between the two studies.

Previous studies from our group investigating the ‘intestinal brake’ by infusing macronutrients in the ileum have repeatedly shown that infusion of even low doses of macronutrients results in a significant reduction of food intake, ranging between 64–188 Kcal, corresponding to a percentual decrease of 11.7%–32% of caloric intake during a single meal [5,21]. This indicates a negative feedback mechanism on food intake that arises from nutrient sensing. These data demonstrate that magnitude of the effect of macronutrient infusion on food intake is greater than the effects of infusing tastants. 

Conclusively, studies investigating differences in food intake should be aware that naso-ileal intubation might mask a small effect. Therefore, other delivery options, such as encapsulation, should be considered in the future.

Results of studies investigating the effects of single tastants on food intake, satiation/satiety, and GI peptides are not consistent. An initial strong decrease of hunger with a steep increase thereafter has been observed after administration of a non-caloric sweetener [22]. Ingestion of low caloric sweeteners did not influence energy intake compared with a control condition (intake of water) [23]. Adding an umami tastant to a meal did not affect appetite sensations, but has been shown to result in an increase of subsequent food intake [24]. Recently, increased attention has been given to the effects of bitter substances on satiety and food intake. Intake or infusion of bitter substances (quinine, denatonium benzoate) not only reduced antral motility [25,26] but also increased satiety scores and resulted in a significant decrease in food intake [27]. A possible mechanism explaining the strong aversive effects of bitter tastants is that bitter taste is evolutionarily linked to toxic substances, as has been showed by presenting newborn infants with bitter substances [28]. 

Alleleyn et al. have shown that the inhibition of food intake shows a proximal to the distal gradient, with higher effects observed after distal versus proximal administration of nutrients [8]. Based on our data, such a gradient was not observed for intestinally administered tastants. Intestinal taste receptor expression varies for various taste receptors, where some taste receptors are more profound proximally in the GI-tract, while expression of other taste receptors is higher in the more distal intestine [14,15,16,17]. 

We thought the proximal to distal gradient found for macronutrient infusion might be operable for taste receptor activation, which was clearly not the case. It is possible that taste receptors inhibit food intake in a different fashion than macronutrients. For instance, it has been speculated that taste receptors function by sensing the type of food (i.e., sweet for carbohydrates, umami for amino acids, and bitter for toxic substances) [29]. Since bitter tastants are linked to toxic substances, another working mechanism for bitter tastants could be through an aversive reaction of subsequent food intake. 

From an evolutionary perspective, a more pronounced inhibitory or aversive effect for toxic substances could be expected to occur in the most proximal parts of the GI tract. However, there are no data available with respect to activation of oral (bitter) taste receptors on subsequent food intake. It is therefore unclear, whether activation of more proximal taste receptors will reveal more pronounced effects on food intake and satiation/satiety. Consequently, further studies are needed to investigate whether more proximal activation of taste receptors results in a stronger decrease in food intake.

Published data on the role of GI peptides in the regulation of food intake after administration of tastants are not in line. Van Avesaat et al. found a clear effect of intraduodenal administration of tastants on food intake that was not accompanied by changes in GLP-1- or PYY level [18]. Other studies, however, did show a decrease in systemic ghrelin- and motilin levels [25,26] and an increase in systemic CCK levels [27] after administration of a bitter tastant.

A limitation of our study is that the wash-out period consisted of only one day. Prolonging the wash-out period over one day would have resulted in a longer period of naso-ileal catheter intubation increasing the discomfort to our volunteers. No interaction effect between intervention and test day was found on food intake, satiety scores or GI-complaints, indicating that no carry-over effect was present.

Another limitation of the present study was the absence of systemic GI hormone measurements. This would have provided a complete analysis of the effects of intestinal tastant administration on eating behavior. However, van Avesaat et al. showed a decrease in food intake and an increase in satiety scores, which was not accompanied by changes in systemic GI hormone levels [18]. Therefore, no systemic GI hormone measurement was conducted in the present study.

It has to be noted that the ideal duration of administration of the intervention and of the timing between intervention and serving the *ad libitum* meal is unknown. We employed a design similar to that of van Avesaat et al. based on their positive results [18]. Future research protocols should consider these factors. 

Studies investigating the effects of tastants on food intake up to now focus on only acute effects in a single *ad libitum* meal. It is not known whether repetitive or chronic administration of tastants will lead to other results. More data are needed on the long-term effects of tastants, especially on daily energy intake.

## Figures and Tables

**Figure 1 nutrients-11-00472-f001:**
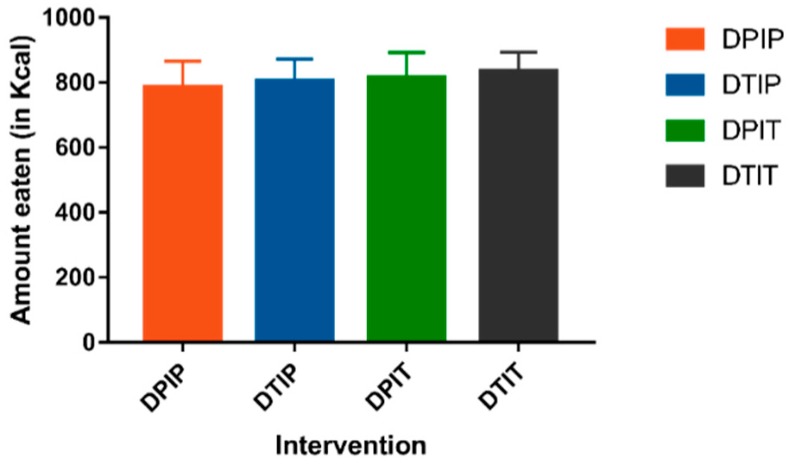
The amount eaten in Kcal (mean + SEM) 15 min after cessation of the infusion of placebo both intraduodenal and intraileal (DPIP), tastants intraduodenal and placebo intraileal (DTIP), placebo intraduodenal and tastants intraileal (DPIT), and tastants both intraduodenal and intraileal (DTIT). Based on a linear mixed model, no difference in food intake was observed between the conditions (*p* = 0.59).

**Figure 2 nutrients-11-00472-f002:**
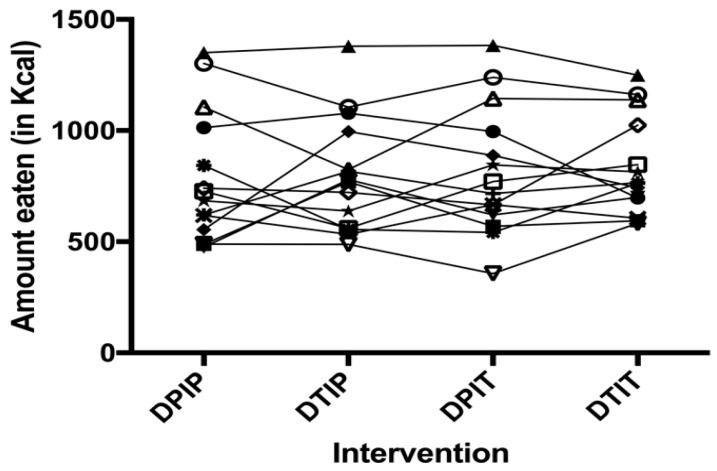
An individual representation per subject of amount eaten in Kcal 15 min after cessation of the infusion of placebo both intraduodenal and intraileal (DPIP), tastants intraduodenal and placebo intraileal (DTIP), placebo intraduodenal and tastants intraileal (DPIT), and tastants both intraduodenal and intraileal (DTIT). Treatment order was randomized for each subject. Each line with a unique symbol represents an individual subject. Based on a linear mixed model, no difference in food intake was observed between the conditions (*p* = 0.59).

**Figure 3 nutrients-11-00472-f003:**
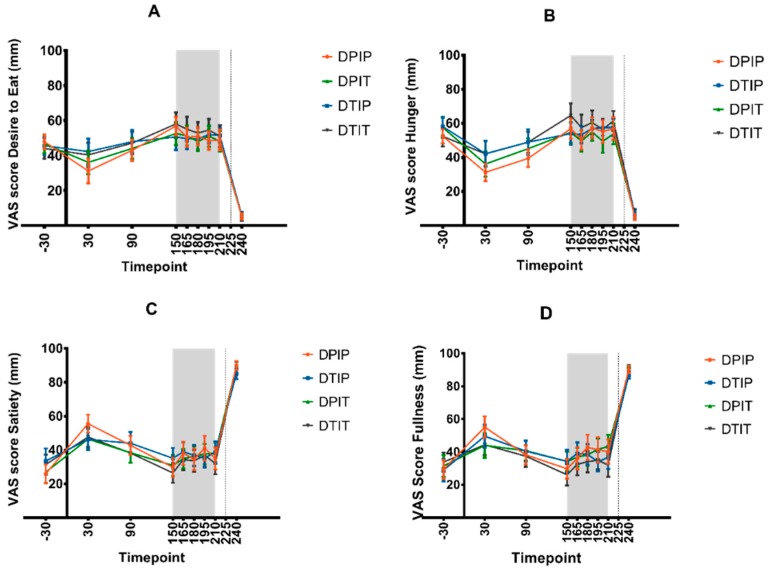
VAS scores for desire to eat (**A**), hunger (**B**), satiety (**C**), and fullness (**D**) (mean + SEM) before, during, and after the infusion of placebo both intraduodenal and intraileal (DPIP), tastants intraduodenal and placebo intraileal (DTIP), placebo intraduodenal and tastants intraileal (DPIT), and tastants both intraduodenal and intraileal (DTIT). VAS scores were measured at t = −30, 30, 90, 150, 165, 180, 195, 210, and 240 min. No VAS scores were taken at t = 225 min. At t = 0 min, subjects received a standardized breakfast, infusion of mixtures was performed from t = 150 until t = 210 min, and *ad libitum* test meal was presented at t = 225. Based on a linear mixed model of mean scores and area under the curve (AUC_150–210_), no differences in desire to eat, hunger, satiety, and fullness were observed between the various conditions.

**Figure 4 nutrients-11-00472-f004:**
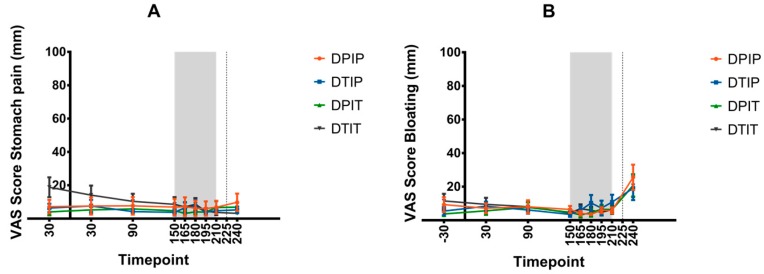
VAS scores for stomach pain (**A**), bloating (**B**), and nausea (**C**) (mean + SEM) before, during, and after the infusion of placebo both intraduodenal and intraileal (DPIP), tastants intraduodenal and placebo intraileal (DTIP), placebo intraduodenal and tastants intraileal (DPIT), and tastants both intraduodenal and intraileal (DTIT). t = −30, 30, 90, 150, 165, 180, 195, 210, and 240 min. No VAS scores were taken at t = 225 min. At t = 0 min, subjects received a standardized breakfast, infusion of mixtures was performed from t = 150 until t = 210 min, and *ad libitum* test meal was presented at t = 225 min. Based on a linear mixed model of mean scores and area under the curve (AUC_150–210_), no differences in stomach pain, bloating, and nausea were observed between the various conditions.

**Table 1 nutrients-11-00472-t001:** A simplified visual representation of the relative expression of taste receptors and gustducin throughout the human GI-tract.

	Stomach	Duodenum	Jejunum	Ileum	Colon
TAS1R1 (Bezencon et al. [14])	++	+	++	+	+/−
TAS1R2 (Bezencon et al. [14])	−	++	+	+/−	+
TAS1R2 (Young et al. [15])	−− ^$^	+	++ ^#^	N/A	N/A
TAS1R3 (Bezencon et al. [14])	+	++	++	+	+
TAS1R3 (van der Wielen et al. [16])	N/A	+	+	+	+
TAS1R3 (Young et al. [15])	+ ^$^	++	++ ^#^	N/A	N/A
TAS2R102–TAS2R144 (Gu et al. [17]) *	N/A	+	+	+	N/A
Gustducin (Bezencon et al. [14])	−−	++	++	+	−
Gustducin (Young et al. [15])	− ^$^	+	++ ^#^	N/A	N/A

Expression levels are relative to each other and a simplified visual representation with ++ indicating very high expression, + indicating high expression, +/− indicating medium expression, − indicating low expression, and −− indicating very low expression. ^$^ Young et al. displayed the stomach as fundus, body, and antrum. For details, please refer to Young et al. [15]. ^#^ Young et al. displayed jejunum as proximal jejunum and distal jejunum. For details, please refer to Young et al. [15]. N/A: not available. * T2R family is expressed throughout the entire small intestine in a comparable fashion with some subtypes more abundant proximally and some distally. For details, please refer to Gu et al. [17].

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
