# Peer review of "Intraintestinal Delivery of Tastants Using a Naso-Duodenal-Ileal Catheter Does Not Influence Food Intake or Satiety"

_nutrients, 2019, doi:10.3390/nu11020472_

Reviewer 1 Report

The current researchers investigated the impact of tastant exposure on food intake and satiety surrounding a single meal. This research builds on prior work  in an attempt to clarify if tastant exposure creates differing impacts by the exposure location. Additional clarity surrounding rationale for this study, as well as inclusion of gut hormone measurement would have made this study more robust and allowed the reader to form more of a conclusion surrounding this research.

Introduction

Line 43: listing original articles would be more useful than a review (reference 9)

Line 44: mention of where taste receptors are located in the GI tract would help contribute to the rationale for conducting the study.

Line 60: It is not clear why you hypothesize that tastant delivery to the ileum would decrease food intake and increase satiation compared to delivery to the duodenum as you state in your discussion (line 239) that evolutionarily one would expect taste receptor expression to be higher proximally.

Methods

Line 125: Timing of the meal versus the infusion should be stated in the methods as done in figure legends.

Line 130: Timing of satiety measurements should be stated in relation to meal time to provide clarity

Results

Line 158: Listing the treatments in the same order as the figure would make the data easier to interpret

Figures 2 and 3: If there was more clarity around when the VAS were performed, it would be easier to see that there is no data point at t = 225 and rather is just when they received their test meal. Otherwise the figures are too small to clearly make this out, bringing into question the trajectory of the change at time 210, 225, and 240.

Discussion

First paragraph content could be condensed into one summary sentence.

Line 204: 64 kcal may have been statistically significant, but not necessarily clinically significant and thus differences in catheter use may or may not have accounted for differences between the studies.

Line 235: More detail about the taste receptor expression in the proximal vs distal small intestine would help provide a rationale for focusing on the distal small intestine in the current study.

Line 239: Again, I’m not sure how this supports your rationale for focusing on the impact of tastants to the distal small intestine.

Line 251: Clarify that there was no interaction effect between intervention and test day for food intake as well.

Limitations: It is not explained why the researchers did not measure GLP-1, PYY, or CCK. They heavily modeled their study around Avesaat et al but yet did not try to replicate this outcome to compare appetite hormone levels between tastant exposure to the ileum vs the duodenum. This would have provided a more robust, quantitative view of potential differences between the impact of tastant exposure to differing parts of the small intestine.

Line 259: I am not sure that including reference 30 provides additional important information. As you eluded, 2 weeks is not long enough to assess changes in body weight.

Conclusion

This paragraph is somewhat repetitive without any overall conclusion of the study included.

Author Response

Reviewer 1

General comments
The current researchers investigated the impact of tastant exposure on food intake and satiety surrounding a single meal. This research builds on prior work in an attempt to clarify if tastant exposure creates differing impacts by the exposure location. Additional clarity surrounding rationale for this study, as well as inclusion of gut hormone measurement would have made this study more robust and allowed the reader to form more of a conclusion surrounding this research.

Response: we thank the reviewer for the kind words and for providing us with suggestions in order to improve the quality of the manuscript. We understand the need for more clarification on the rationale for the study and tried to improve this. We fully agree that measurement of systemic gut hormone levels could have provided a more complete view and also understanding of the effects of intestinal tastant administration on eating behavior.  We tried to implement as much of the suggestions as possible in order to improve our manuscript.

Specific points

 Introduction

Line 43: listing original articles would be more useful than a review (reference 9)

Response: Reference 9 did nicely summarize what we intended to postulate. However, we agree that original articles are more useful and added them as references.

Line 44: mention of where taste receptors are located in the GI tract would help contribute to the rationale for conducting the study.

Response: Due to word limit, we initially restricted the discussion of this item. However, we agree with the reviewer that this information is relevant in terms of justifying the study design and in particular with relevance to infusion in the distal small intestine. The taste receptors can be found throughout the entire GI-tract: mouth, stomach, small intestine and large intestine. Since it differs per taste receptor where the expression is highest in the gut, we included a table to highlight the differential expression of the taste receptors that we believe are relevant to the substances investigated in this study.

Line 60: It is not clear why you hypothesize that tastant delivery to the ileum would decrease food intake and increase satiation compared to delivery to the duodenum as you state in your discussion (line 239) that evolutionarily one would expect taste receptor expression to be higher proximally.

Response: We thank the reviewer for pointing this out. It is apparent that the proximal small intestine generally is exposed to higher amounts of food-derived substances including tastants. However, not all taste receptors are expressed to the highest degree in the proximal small intestine. We hypothesized therefore, based on the concept of the ileal brake, that when the distal small intestine is exposed to tastants, the ‘brake’ on food intake would be even more pronounced as sensing food seems to be one of the functions of intestinal taste receptors. This does not appear to be the case. We elaborated more on this in both our introduction and discussion.

Methods

Line 125: Timing of the meal versus the infusion should be stated in the methods as done in figure legends.

Response: timing of the meal in relation to the infusion has been added to the methods.

Line 130: Timing of satiety measurements should be stated in relation to meal time to provide clarity

Response: timepoints of VAS measurement were added to clarify the VAS measurements in relation to meal time. Moreover, in paragraph ‘2.5 protocol’ the timepoints of infusion and timepoint of the ad libitum pasta meal were added in order to better clarify the protocol.

Results

Line 158: Listing the treatments in the same order as the figure would make the data easier to interpret

Response: we listed the treatments in the same order as the figure in order to make the data easier to interpret. We fully agree that this improves readability.

Figures 2 and 3: If there was more clarity around when the VAS were performed, it would be easier to see that there is no data point at t = 225 and rather is just when they received their test meal. Otherwise the figures are too small to clearly make this out, bringing into question the trajectory of the change at time 210, 225, and 240.

Repsonse: we agree that this is unclear in a small figure. As stated above, in the methods we added the timepoints of the VAS scores and the test meal. Moreover, we added text stating that t=225 is solely the timepoint that the test meal was provided in the figure legends in order to clarify this.

Discussion

First paragraph content could be condensed into one summary sentence.

Response: we put the first paragraph content into one summary sentence. We agree that it does not have to be as elaborate as previously stated.

Line 204: 64 kcal may have been statistically significant, but not necessarily clinically significant and thus differences in catheter use may or may not have accounted for differences between the studies.

Response: we agree that this is a small difference which in itself might not be clinically significant. However, we do consider that this is a difference in food intake that can be clinically significant if this could be repeated over the course of the day (breakfast, lunch, dinner). This was added to the discussion.

Line 235: More detail about the taste receptor expression in the proximal vs distal small intestine would help provide a rationale for focusing on the distal small intestine in the current study.

Response: taste receptors are found throughout the entire gastro-intestinal tract. It differs per taste receptor whether expression is more pronounced proximally or distally in the gut. We refer also to the comment on the introduction.

Line 239: Again, I’m not sure how this supports your rationale for focusing on the impact of tastants to the distal small intestine.

Response: When starting this study we expected the mechanism to be comparable to the ileal break, where a more pronounced effect on food intake was found when infusing nutrients into the distal parts of the GI-tract. We thought that the presence of macronutrients and their degradation products more distal accounted for a larger effect due to unprocessed food being present in the distal gut, which could mean that the gut senses an overload of unprocessed food indicating that one is full and inhibiting further intake. Since taste receptors in the GI tract seem to act via a mechanism to sense food being present, we thought it might show the same proximal to distal gradient. This part was added to this paper as an advice for the focus of further studies. This was elaborated more clearly by adding this in a sentence and by introducing this more clearly.

Line 251: Clarify that there was no interaction effect between intervention and test day for food intake as well.

Response: added to the results for food intake that no interaction between intervention and test day was found and further elaboration on this in the discussion.

Limitations: It is not explained why the researchers did not measure GLP-1, PYY, or CCK. They heavily modeled their study around Avesaat et al but yet did not try to replicate this outcome to compare appetite hormone levels between tastant exposure to the ileum vs the duodenum. This would have provided a more robust, quantitative view of potential differences between the impact of tastant exposure to differing parts of the small intestine.

Response: we do agree that measurement of satiety hormones would provide a more robust and quantitative view of the potential differences following tastant exposure to the different parts of the small intestine. Van Avesaat et al. found a statistically significant effect on food intake and satiety scores that was not accompanied by changes in GI hormone levels. In the current study our aim was to explore whether a location dependent effect on eating behavior is present, not to search for underlying mechanisms. Therefore, we chose not to include this measurement in the present study. We are aware that this is a limitation and based on your recommendations added this to our discussion.

Line 259: I am not sure that including reference 30 provides additional important information. As you eluded, 2 weeks is not long enough to assess changes in body weight.

Response: we chose to delete the reference based on your recommendation. However we do think the rest of the paragraph is important, since it underlines the need for long-term intervention studies on this subject.

Conclusion

This paragraph is somewhat repetitive without any overall conclusion of the study included.

Response: we agree. Due to repetitiveness of the paragraph, we deleted this paragraph entirely.

Reviewer 2 Report

The authors examined the impact of duodenal and ileal infusion of tastants on food intake and satiety in human subjects. There was no apparent effect of either duodenal or ileal infusion. No rational explanations are provided for the lack of effect and there is excessive speculation on why no effect was observed. For example, it is suggested that prolonged intubation may have impacted the findings in the present study. But this was apparently not an issue in previous work by the authors which is cited. There are just too many unanswered questions in study design, choice of tastants, and execution, for the body of work to be considered suitable for publication. The study does not advance knowledge on food intake and satiety.  

Author Response

Reviewer 2

General comments

The authors examined the impact of duodenal and ileal infusion of tastants on food intake and satiety in human subjects. There was no apparent effect of either duodenal or ileal infusion. No rational explanations are provided for the lack of effect and there is excessive speculation on why no effect was observed. For example, it is suggested that prolonged intubation may have impacted the findings in the present study. But this was apparently not an issue in previous work by the authors which is cited. There are just too many unanswered questions in study design, choice of tastants, and execution, for the body of work to be considered suitable for publication. The study does not advance knowledge on food intake and satiety.

Response: thank you for your critical appraisal of our study. We are aware that there is a lot of speculation on why we observed no effect. In the earlier work that we cited from van Avesaat et al. there was a nasoduodenal intubation on each test day, but the nasoduodenal catheter was removed before the test meal was presented to the participants. In the other studies cited that did use a prolonged nasoileal intubation, macronutrients were infused instead of tastants. We attempted to explain that the effects of macronutrients are generally larger than the effects found after gastrointestinal tastant delivery. This could be one of the reasons that an effect of macronutrients was not masked by discomfort and negative effect of prolonged intubation.

As described in the introduction, we chose these tastants because bitter, sweet and umami are sensed by various families of G-coupled protein receptors. We chose for the same amount and same kind of tastants as van Avesaat et al., because we had previous experience with these tastants. Salty and sour taste receptors are ion-channels and since ion-channels have various functions in the GI-tract we chose not to add them to the mixture.

We improved our manuscript based on the other reviewer’s comments and hope that based on our response you would reconsider the suitability of our manuscript for publication.

Reviewer 3 Report

This is a well-designed study which yielded results that did not fit with the hypothesis. Nevertheless, the results are an important contribution to the literature, and the authors should be confident in presenting these findings.  

The methodology is clearly described, and the interpretation and discussion of results is appropriate.

Further value might be added with regards to the following:

- Clarify the composition and nutritional value (if any) of the tastants used.

- What was the justification for the infusion rate of the tastants?

These details will be useful to consider when making comparisons to previous studies involving macronutrients.

- Consider showing individual responses to treatments rather than just the mean responses? With a study of this size a graphical representation for each subject could visually give an indication of how individualised this response is.

-In the discussion there is some speculation around whether or not the difference in intubation technique between the present study and previous studies is responsible for inconsistent outcomes. This could be an important consideration for future study protocols, and it would be helpful if the authors could be more conclusive about this aspect of the methodology. 

-The explanation and discussion around mechanistic basis for tasters interacting with duodenal and ileal receptors could be expanded. 

- It would be interesting to know if the composition of the breakfast meal would have an impact on subsequent sensitivity of these receptors. 

Author Response

Reviewer 3

General comments

This is a well-designed study which yielded results that did not fit with the hypothesis. Nevertheless, the results are an important contribution to the literature, and the authors should be confident in presenting these findings. 

The methodology is clearly described, and the interpretation and discussion of results is appropriate.

Response: we thank the reviewer for the kind words and suggestions. We improved our manuscript according to the reviewer’s suggestions.

Specific points

Further value might be added with regards to the following:

- Clarify the composition and nutritional value (if any) of the tastants used. 

Response: the tastants themselves do not yield any nutritional value. This was added to the methods section.

- What was the justification for the infusion rate of the tastants? 

Response: The infusion rate was similar to that of van Avesaat et al. and mimicks slow influx from stomach to duodenum and slow transit through the gut in ileal infusion. This was added to the manuscript in the methods section.

These details will be useful to consider when making comparisons to previous studies involving macronutrients.

- Consider showing individual responses to treatments rather than just the mean responses? With a study of this size a graphical representation for each subject could visually give an indication of how individualised this response is.

 Response: We chose to present the mean responses in order to conform to the usual way of presenting this kind of results. Below is an example of how individualized data would look like. We do not think this provides more insight in the response, as the bar graph gives a nice and structured view and clearly shows that there is no treatment effect. Moreover, graphs as below tend to suggest a certain order in treatment, whereas our treatments were completely randomized. In case the reviewers and the editors find the presentation in this form relevant, we have added this as a supplementary figure. I see the figure does not upload well in this text box, so I would like to refer you to the cover letter and uploaded supplementary figure.

-In the discussion there is some speculation around whether or not the difference in intubation technique between the present study and previous studies is responsible for inconsistent outcomes. This could be an important consideration for future study protocols, and it would be helpful if the authors could be more conclusive about this aspect of the methodology.

Response: We elaborated more on this assumption by giving the advice for further protocols to be aware of this possible masking of small effects and to consider other delivery techniques.

-The explanation and discussion around mechanistic basis for tasters interacting with duodenal and ileal receptors could be expanded.

Response: We thought that sensing of the presence of food was one of the main functions of tastants. As described in the introduction, tastants are known to be able to result in the release of GI hormones in several in vivo and in vitro studies. However, as now described in the discussion van Avesaat et al. found the effects of tastants on food intake and satiety may not accompanied by any changes in GI hormones. Van Avesaat et al. further speculate on the possible vagal activation and paracrine effects of tastants. Added to our discussion now is a possible aversive way in which the taste receptors (mainly for bitter components) could exert their mechanism. This could be a reason why no effect was found more distally in the GI-tract. Therefore, we advise further research protocols to focus on proximal taste receptor activation.

- It would be interesting to know if the composition of the breakfast meal would have an impact on subsequent sensitivity of these receptors.

Response: this would indeed be very interesting. Unfortunately, our study was not designed  to investigate this question. This would allow a separate research question in its own and was beyond the scope of the current study. However, it must be noted that van Avesaat et al. used the same breakfast with the same composition. Thus, this is not the cause for our different findings.

Round  2

Reviewer 2 Report

I have no further comments on the manuscript.

Author Response

General comments

I have no further comments on the manuscript.

Response: we want to thank the reviewer for rereading our manuscript critically and for accepting our revisions.